# Relationship between Psoriasis and Prevalence of Cardiovascular Disease in 88 Japanese Patients

**DOI:** 10.3390/jcm10163640

**Published:** 2021-08-17

**Authors:** Fumikazu Yamazaki, Kazuya Takehana, Akihiro Tanaka, Yonsu Son, Yoshio Ozaki, Hideaki Tanizaki

**Affiliations:** 1Departments of Dermatology, Kansai Medical University, Osaka 573-1010, Japan; tanizakh@hirakata.kmu.ac.jp; 2Psoriasis Center, Kansai Medical University, Osaka 573-1191, Japan; takehana@hirakata.kmu.ac.jp (K.T.); tanakaa@hirakata.kmu.ac.jp (A.T.); sony@hirakata.kmu.ac.jp (Y.S.); ozakiy@takii.kmu.ac.jp (Y.O.); 3Division of Cardiology, Departments of Medicine II, Kansai Medical University, Osaka 573-1010, Japan; 4Division of Rheumatology, Departments of Medicine I, Kansai Medical University, Osaka 573-1010, Japan

**Keywords:** ABI (ankle–brachial pressure index), CVD (cardiovascular disease), CCTA (coronary artery and cardiac computed tomography), Japanese, psoriasis

## Abstract

Psoriasis is a systemic inflammatory disease known to affect survival in the presence of cerebral or cardiovascular comorbidities. However, no clear guidelines have been defined regarding the extent of vascular lesion testing that should be performed in patients with psoriasis. We therefore performed coronary computed tomography angiography (CCTA) in 88 Japanese patients with psoriasis who visited Kansai Medical University Hospital between 2015 and 2019 and determined the ankle–brachial pressure index (ABI) for 44 of these patients. CCTA abnormalities were found in 39 of the 88 patients, and a need for treatment was identified in 14 patients. The prevalence of cardiovascular lesions in these patients was 15.9%, significantly higher than that in the healthy Japanese population (6.38% according to the Suita Study). In the 44 patients with results for both ABI and CCTA, the rates of CCTA vascular lesions were significantly higher in cases with ABIs indicating hard vessels or above than in cases with supple, normal, or slightly stiff vessels. This is the first report to show a correlation between CCTA and ABI in psoriasis patients. ABI was considered useful as a preliminary test before CCTA. The univariate analysis of the abnormal and normal CCTA groups showed that the prevalence differed significantly among patients with psoriatic arthritis, erythrodermic psoriasis, older age, pre-existing conditions, drinking, and hypertension. The multivariate analysis showed correlations with arthritic or erythrodermic psoriasis.

## 1. Introduction

Psoriasis is a systemic inflammatory disease that is known to cause arthritis and uveitis [1]. A recent cohort study from the United Kingdom found that patients with moderate to severe psoriasis have lifespans around 6 years shorter than those of healthy individuals. This has been postulated to be attributable to cardiovascular disease (CVD) caused by inflammation, such as myocardial infarction and cerebral infarction [2]. According to World Health Organization statistics, the prevalence of coronary artery disease and vascular lesions is greatest among individuals with diabetes and hypertension, although the mortality rate is highest among individuals with psoriasis [3]. This may be because diabetes and hypertension are commonly recognized as significant risk factors for vascular lesions, so patients and physicians proactively seek and provide/receive treatment. By contrast, because psoriasis is not generally recognized as a risk factor for vascular lesions, these patients are often overlooked. Currently, no clear consensus has been reached regarding how much testing of vascular lesions should be performed in patients with psoriasis, and data for Japanese populations are lacking. We therefore performed pre-treatment coronary computed tomography angiography (CCTA) and determined the ankle–brachial pressure index (ABI) in patients with moderate to severe psoriasis with assistance from the cardiology department to investigate the prevalence of vascular disorders and associated factors. The objectives of this analysis were to investigate the usefulness of CCTA in identifying vascular lesions and to clarify the associations between these lesions and the psoriasis type and lifestyle in patients with moderate to severe psoriasis.

## 2. Materials and Methods

CCTA was performed in 88 Japanese patients with psoriasis who visited Kansai Medical University Hospital between 2015 and 2019, and the ABI was determined for 44 of these 88 patients in Table 1 (Appendix A). Diagnoses of psoriasis were made by a dermatologist using examination and biopsy, and diagnoses of psoriatic arthritis (PsA) were made by a rheumatologist; all diagnoses met the CASPAR diagnostic criteria. With vascular lesions as the response variable and age, sex, psoriasis type, pre-treatment psoriasis area and severity index (PASI) score, body mass index (BMI), complications, C-reactive protein (CRP), blood pressure (systolic blood pressure (SBP])diastolic blood pressure (DBP)), family history of psoriasis (FHP), familial history of cardiovascular disease (FHC), hyperlipidemia, diabetes, smoking, and alcohol consumption as explanatory variables, we conducted univariate logistic regression analyses to determine the odds ratio for each explanatory variable. Moreover, explanatory variables showing a potential association (*p* < 0.15 in the univariate analyses) were entered into a multivariate model to determine the adjusted odds ratios. Analyses were conducted to determine whether a difference was present between groups with and without vascular lesions. Fisher’s exact test was performed for categorical variables, and a t-test was performed for quantitative variables. For the ABI, measurement results were evaluated using the following 5-point scale: 1, “Supple” (ABI > 1.1); 2, ”Normal” (ABI < 1–0.9); 3, “Slightly stiff” (ABI < 0.9–0.7); 4, “Hard” (ABI < 0.7–0.5); and 5, “Clogged” (ABI < 0.5). The categories were compared with the presence of CCTA lesions using Fisher’s exact test. The CCTA and ABI tests were performed only in patients with psoriasis who visited Kansai Medical University and gave their consent.

## 3. Results

Of the 88 Japanese patients with psoriasis, 39 presented with cardiovascular abnormalities, 25 (28.4%) required cardiovascular testing, and 14 (15.9%) needed treatment (psoriatic arthritis (PsA), *n* = 9; psoriasis vulgaris (PV), *n* = 3; generalized pustular psoriasis (GPP), *n* = 1; erythrodermic psoriasis (EP), *n* = 1) (Table 1 and Appendix A). This prevalence was significantly higher than the 6.38% prevalence of coronary artery disease in the control group of the Suita City study (*p* < 0.001) [4].

### 3.1. Statistical Study

#### 3.1.1. Univariate Analysis

In the univariate analyses, with regard to disease type, the prevalence was significantly greater in patients with PsA (odds ratio, 5.16; *p* = 0.001) and EP (odds ratio, 6.998; *p* = 0.089) than in those with PV. In addition, there was a significant increase in elderly (odds ratio: 1.069, *p* = 0.0005), SBP (odds ratio: 1.029, *p* = 0.0285), complications (odds ratio: 2.697, *p* = 0.043), FHC (odds ratio: 0.117, *p* = 0.0204), and alcohol consumption (odds ratio: 2.388, *p* = 0.0481), (Table 2).

#### 3.1.2. Multivariate Analysis

In the multivariate analysis, the prevalence was higher in patients with PsA (odds ratio, 6.945; *p* = 0.0313) and EP (odds ratio, 41.595; *p* = 0.0207) than in patients with PV (Table 2).

In the results of both the univariate and multivariate analyses, the PASI score may not appear to be proportional to the prevalence of cardiovascular disease in the Japanese population.

In fact, in the statistical analysis of our 88 patients, there was no correlation between the PASI score and the prevalence of CVD (odds ratio, 0.991, *p* = 0.6814) (Table 2). We believe the reason for this is that while some cases, such as EP, had high PASI scores and showed abnormalities on the CCTA, there were also PsA cases with low PASI scores and abnormalities on the CCTA, and these cases were averaged out. 

Risk factors that are highly correlated with CVD in the Japanese population, such as BMI, hyperlipidemia, diabetes, smoking, gender, and high-sensitivity CRP, were not correlated with prevalence in this study, which conflicts with the conventional theory of cardiovascular disease (Table 2). In addition, FHC was negatively correlated with abnormalities in CCTA (odds ratio, 0.117, *p* = 0.0204), suggesting that cardiovascular disease may develop independently of FHC.

### 3.2. Cases

The subset of cases in which CCTA was performed is described in the Appendix A.

### 3.3. CCTA and ABI

Among the 44 patients for whom the ABI was determined, the prevalence of abnormalities on CCTA was significantly higher for cases with ABI classified as hard or above than in cases with supple, normal, or slightly stiff results (*p* = 0.0329) (Table 3).

In summary, patients who were more likely to present with abnormalities on CCTA were more likely to have PsA or EP, older age, higher SBP, comorbidities, and regular alcohol consumption. On the other hand, an ABI result of ABI <0.7–0.5 may indicate an abnormality on CCTA. In contrast to previous epidemiological data on CVD, there was no correlation with BMI, high-sensitivity CRP, or smoking. There was also no correlation between the prevalence of abnormalities on the CCTA and the PASI score, which indicates the severity of psoriasis.

## 4. Discussion

Psoriasis is known to be associated with obesity [5]. Psoriasis has been associated with a high incidence of CVD and its complications, but it was previously thought to be an incidental consequence of obesity itself or poor diet [5]. Recently, however, cytokine analysis of psoriasis patients has led to the mainstream view that psoriasis is a systemic inflammatory disease and that cytokines have cardiovascular effects [6]. The cytokines currently thought to be associated with psoriasis include tumor necrosis factor (TNF)-α, interleukin (IL)-17, and IL-23, but TNF-α is thought to be the cytokine with the greatest impact on the cardiovascular system [7].

Elevations in TNF-α increase adipokines and vascular endothelial impairment, and decreases in glucose tolerance result in the progression of arteriosclerosis, thereby inducing myocardial infarction. This mechanism of progression is called the “psoriatic march” [8].

Although patients with moderate to severe psoriasis live about 6 years less than healthy individuals, the use of TNF-α inhibitors resulted in a longer lifespan than that of the healthy population [9]. This finding supports the existence of an effect of TNF-α on the cardiovascular system.

Hence, in psoriasis patients with cardiovascular risk, the proactive use of TNF-α inhibitors appears to be warranted. To accomplish this, effective tools for identifying CVD in patients with psoriasis is necessary.

In the first study to use CCTA in Western countries, patients with severe psoriasis displayed a greater frequency of CVD with an abnormality rate of approximately 40% [10]. Furthermore, this risk was considered proportional to the severity of psoriatic skin eruptions [11] and the duration of psoriasis [12]. We therefore investigated whether the same held true in the Japanese population. Our statistical analyses could not investigate the duration of disease, but we found that Japanese patients with psoriasis did not necessarily display proportionally greater cardiovascular risk with increasing PASI scores. This may have been due to differences in ethnicity or diet.

On the other hand, in Western countries, PsA is associated with a greater rate of CVD complications [13]. These findings were similar to our findings in the Japanese population.

Older, hypertensive psoriasis patients showed a significantly greater prevalence of CVD than other groups, similar to the general trend. High-sensitivity CRP was not significantly associated with the detection of cardiovascular lesions in this study because some vascular lesions may have been detected at a stage before the appearance of dysfunction. The development of a more sensitive blood test is thus desired.

Smoking is a common risk factor for CVD in the Japanese population. However, vascular lesions were detected in the non-smoking group in the present study, indicating that Japanese patients with psoriasis who are non-smokers also need to be cautious. This study showed a trend toward higher CCTA abnormalities in the group with regular alcohol consumption. Some have questioned the observational data showing an association between alcohol and CVD for methodological reasons. Most often pointed out are the facts that lifelong abstainers and former drinkers are lumped together in some analyses and that baseline CVD is poorly controlled for in some studies, but it is generally believed that moderate alcohol consumption reduces the risk of CVD [14]. In the present study, there was a correlation between alcohol consumption and abnormalities of CVD, but since we only asked about the presence of drinking habits, we could not examine the correlation with the amount of alcohol consumed and other factors. This is one of the issues that need to be analyzed in the future.

CCTA examination was performed in Japanese patients with psoriasis, with approximately 30% exhibiting abnormalities and about 15% needing treatment. This demonstrated that comorbid CVD is very common, similar to data from Western countries and supporting the utility of CCTA.

Although CCTA has proven useful as a tool for identifying CVD in psoriatic patients, performing this imaging in all psoriatic patients is unlikely to be feasible. This study investigated the possibility of performing non-invasive examinations in psoriasis patients, with additional CCTA performed once vascular lesions are suspected, in order to more efficiently search for vascular lesions.

The ABI is a marker of general arterial stiffness and predicts future cardiovascular events [15]. In addition to the ABI, other non-invasive tests include intima–media thickness, flow-mediated dilation, nitrate-mediated dilation, pulse wave velocity, and augmentation index. Each of these has been investigated as a tool for identifying vascular disease in patients with psoriasis [15]. The ABI is a non-invasive test that can be used to detect vascular disease in patients with psoriasis. However, these tests, including the ABI, are not reproducible on their own and cannot rule out other contributing factors [16].

We performed statistical analyses for 44 patients, combining ABI and CCTA, as these are the simple methods to measure limb pressure. As a result, we detected abnormalities on CCTA significantly more frequently in psoriasis patients with ABI results Hard (4) <0.9–0.7 or above than in patients with results of slightly stiff or below (*p* = 0.0329) (Table 3). In other words, the ABI appears to be useful as a preliminary test before performing CCTA. This is the first report to suggest that the combination of the ABI and CCTA may prove useful in detecting vascular lesions in psoriasis patients.

On the other hand, the combination of CCTA with other non-invasive tests (intima—media thickness, flow-mediated dilation, nitric acid-mediated dilation, pulse wave velocity, augmentation factor, etc.) may be useful as a preliminary test for performing similar CCTA and needs further investigation.

CCTA was useful for detecting cardiovascular lesions in patients with psoriasis, but it cannot be performed in patients with decreased renal function. In such cases, other methods of detection, such as myocardial scintigraphy, may need to be considered.

## 5. Conclusions

The use of CCTA to search for CVD in Japanese psoriasis patients seems appropriate, but caution should be exercised even if, unlike Caucasians, Japanese patients have low PASI scores and BMIs. The ABI was also found to be useful as a preliminary test (pretest) for CCTA. Dermatology appears to have a substantial role to play in examining patients with psoriasis as a systemic disease in order to detect cardiovascular lesions at an early stage.

## Figures and Tables

**Table 1 jcm-10-03640-t001:** Characteristics of psoriasis patients who underwent CCTA at Kansai Medical University.

Analysis Items	Numbers of Patients	% & *p*-Value *
All	88	
Male	62	
Female	26	
PV	45	
PsA	33	
EP	4	
GPP	6	
with ABI	44	
Cardiovascular abnormalities in CCTA		
Yes	39	44.3
No	49	55.7
Need for cardiovascular examination		
Yes	25	28.4 * *p*-value < 0.001
No	63	71.6
Need for cardiovascular treatment		
Yes	14	15.9 * *p*-value < 0.001
PV	3	
PsA	9	
EP	1	
GPP	1	
No	74	84.1
Incidence of cardiovascular disorders in healthy individuals in Suita City (Suita Study)		6.38
Age		
33–83 (Mean: 59.5)		
Mean PASI: 12.1		
Mean BMI: 25.0		
* T-Test: Need for cardiovascular examination and Need for cardiovascular treatment with two-tailed test of Suita Study		

ABI: ankle-brachial pressure index, BMI: body mass index, CCTA: coronary computed tomography angiography, EP: erythrodermic psoriasis, GPP: generalized pustular psoriasis, PASI: psoriasis area and severity index, PsA: psoriasis arthritis, PsV: psoriasis vulugaris.

**Table 2 jcm-10-03640-t002:** Four types of psoriasis/LOGISTIC regression analysis.

Analysis Items	Types	Odds Ratio	95% Cl	*p*-Value **
(univariate analysis )				
predictor variable *				
Age		1.069	(1.030, 1.110)	0.0005
Sex (female)		1.729	(0.671, 4.454)	0.2571
Type (PsV)	PsA	5.160	(1.941, 13.718)	0.0010
	GPP	5.400	(0.892, 32.703)	0.0665
	EP	6.998	(0.743, 65.889)	0.0890
PASI		0.991	(0.948, 1.036)	0.6814
BMI		0.933	(0.839, 1.038)	0.2018
complications		2.697	(1.026, 7.093)	0.0443
CRP		1.052	(0.648, 1.708)	0.8364
BP	SBP	1.029	(1.003, 1.056)	0.0285
	DBP	1.015	(0.985, 1.046)	0.3175
FHC		0.117	(0.019, 0.718)	0.0204
Hyperlipidemia		1.338	(0.575, 3.114)	0.4993
Diabetes		1.806	(0.570, 5.724)	0.3151
Smoking		0.889	(0.373, 2.124)	0.7919
Drinkng		2.388	(1.007, 5.660)	0.0481
(multivariate analysis)				
predictor variable *		Odds ratio	95% Cl	*p*-value
Age		1.037	(0.979, 1.098)	0.2172
Type (PsV)	PsA	6.945	(1.190, 40.530)	0.0313
	GPP	3.541	(0.251, 49.989)	0.3493
	EP	41.595	(1.769, 977.972)	0.0207
complications		0.596	(0.090, 3.968)	0.5930
BP	SBP	1.047	(0.993, 1.104)	0.0896
FHC		0.060	(0.003, 1.195)	0.0653
Drinkng		3.084	(0.671, 14.185)	0.1480

* ( )Category with denominator inside; ** logistic regression. BMI: body mass index, BP: blood pressure, CCTA: coronary computed tomography angiography, DBP: diastolic blood pressure, EP: erythrodermic psoriasis, FHC: familial history of cardiovascular disease, GPP: generalized pustular psoriasis, PASI: psoriasis area and severity index, PsA: psoriasis arthritis, PsV: psoriasis vulugaris. SBP: systolic blood pressure.

**Table 3 jcm-10-03640-t003:** Comparison of ABI assessments by the prevalence of CCTA abnormalities.

CCTA ^a^	Cases	Supple (1) > 1.1	Normal (2) < 1–0.9	Slightly stiff (3) < 0.9–0.7	Hard (4) < 0.7–0.5	Clogged (5) < 0.5
-	20	0	12 (60)	2 (10)	5 (25)	1 (5)
+	24	1 (4.2)	4 (16.7)	3 (12.5)	15 (62.5)	1 (5)
**CCTA ^b^**	**Cases**	**(1) + (2)**	**(3) + (4) + (5)**			
-	20	12 (60)	8 (40)			
+	24	12 (20.8)	19 (72.9)			
**CCTA ^c^**	**Cases**	**(1) + (2) + (3)**	**(4) + (5)**			
-	20	14 (70)	6 (30)			
+	24	8 (33.3)	16 (66.7)			

^a^: Wilcoxon 2sample test, *p* = 0.0227; ^b^: Fisher’s exact test, *p* = 0.0128; ^c^: Fisher’s exact test, *p* = 0.0329. CCTA: coronary artery cardiac computed tomography; ABI: ankle–brachial pressure index.

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
