# Peer review of "Relationship between Psoriasis and Prevalence of Cardiovascular Disease in 88 Japanese Patients"

_jcm, 2021, doi:10.3390/jcm10163640_

Round 1
Reviewer 1 Report
This study addresses the important question of whether psoriasis (Ps) patients are at a greater risk of developing cardiovascular diseases (CVD) and suggests clinical factors associated with CVD. Also, the usefulness of ABI as a simple, inexpensive measure to predict CVD risk has been demonstrated. The article is generally well-written and concise. However, there are some suggestions I would like to make for possible improvement:
Introduction
- line 46: Please add a suitable reference for the correlation between Ps and uveitis.
Materials and methods
- For the patients included in this study, how was the diagnosis of Ps made? Was the diagnosis based on clinical impressions or was skin biopsy performed?
- How was the diagnosis of psoriatic arthritis made? Did clinicians use clinical criteria e.g. CASPAR?
- Was CCTA performed for all patients with Ps who visited your hospital? If not, please specify how patients who took CCTA were selected. And what about ABI? How were patients who did ABI tests selected?
Results
- line 85: it seems better to add “cardiovascular” abnormalities for clarity.
- line 94: does PsA patients refer to those with psoriatic arthritis without any skin changes due to Ps? Please specify the type of Ps lesions in PsA patients.
- line 111-112: Please state what treatment was done for acute MI.
- lines 163-164: the sentence “this is the first…” would fit better in the discussion section.
- Individual figure panels should be compiled together (Figure 1a to 1c put together, the same for Figure 2a to 2e and so on) along with the corresponding legends.
- The last figure lacks figure legend.
- Table 1: a more concise table consisting of a summary of demographic characteristics (e.g. age: mean ±SD, sex M:F, etc) could be better than the present Table 1. Overall, the tables could be improved in their presentation: too many grids are present.
Discussion
- lines 243-245: it has been postulated the Ps is a systemic autoinflammatory disease. This could be added, together with appropriate reference, to the first paragraph in the Discussion.
- line 278: Isn’t alcohol consumption an independent risk factor for CVD? If it is not the case for Japanese individuals, please add a relevant reference on this issue.
- line 285: the full stop (.) after CVD should be omitted.
- I agree that ABI is a simple test that could potentially be used for CVD risk screening. But what about other tests such as EKG or Doppler ultrasound? The possible clinical usefulness of other test modalities could be mentioned in the Discussion and may be a subject of future study.
- Nail involvement or nail changes in Ps patients was not evaluated in this study. I would suggest the authors assess the association between CVD and psoriatic nail changes if relevant data are available.
I believe the above comments would be of help in improving the quality of your submitted article.
Author Response
Thank you for reviewing it. I will provide my response below.
Introduction
line 46: Please add a suitable reference for the correlation between Ps and uveitis.
→Corrected and added additional reference 1.
Materials and methods
For the patients included in this study, how was the diagnosis of Ps made? Was the diagnosis based on clinical impressions or was skin biopsy performed?
How was the diagnosis of psoriatic arthritis made? Did clinicians use clinical criteria e.g. CASPAR?
Was CCTA performed for all patients with Ps who visited your hospital? If not, please specify how patients who took CCTA were selected. And what about ABI? How were patients who did ABI tests selected?
→Added evidence of diagnosis in dermatology and rheumatology (CASPAR), and added a statement that CTTA and ABI were performed only on psoriasis patients who gave consent.
Results
line 85: it seems better to add “cardiovascular” abnormalities for clarity.
→Corrected.
line 94: does PsA patients refer to those with psoriatic arthritis without any skin changes due to Ps? Please specify the type of Ps lesions in PsA patients.
→PsA was only available for cases diagnosed in rheumatology, and since skin rash could not be defined, only cases diagnosed in rheumatology and meeting CASPAR were included.
line 111-112: Please state what treatment was done for acute MI.
→I've added it.
lines 163-164: the sentence “this is the first…” would fit better in the discussion section.
→Deleted.
Individual figure panels should be compiled together (Figure 1a to 1c put together, the same for Figure 2a to 2e and so on) along with the corresponding legends.
The last figure lacks figure legend.
→Adjusted the figures and attached the Case introduction as supplementary data.
Table 1: a more concise table consisting of a summary of demographic characteristics (e.g. age: mean ±SD, sex M:F, etc) could be better than the present Table 1. Overall, the tables could be improved in their presentation: too many grids are present.
→Table 1 has been changed for simplicity and clarity. I have decided to attach the original Table 1 as supplemental data.
Discussion
lines 243-245: it has been postulated the Ps is a systemic autoinflammatory disease. This could be added, together with appropriate reference, to the first paragraph in the Discussion.
→Corrected.
line 278: Isn’t alcohol consumption an independent risk factor for CVD? If it is not the case for Japanese individuals, please add a relevant reference on this issue.
→Corrected and added additional reference 14.
line 285: the full stop (.) after CVD should be omitted.
→Corrected.
I agree that ABI is a simple test that could potentially be used for CVD risk screening. But what about other tests such as EKG or Doppler ultrasound? The possible clinical usefulness of other test modalities could be mentioned in the Discussion and may be a subject of future study.
→I've decided to add a note on those points.
Nail involvement or nail changes in Ps patients was not evaluated in this study. I would suggest the authors assess the association between CVD and psoriatic nail changes if relevant data are available.
→Thank you for pointing this out. We will take this point into consideration in the future.
Please find attached the corrected text.

Reviewer 2 Report
This study describes cardivascular comorbodities in a series of 88 patients with psoriasis.
Below my comments:
1) abstract is too long. The authors should separate the various sections providing a summary of the methods, results and conclusions.
2) the methods are not clearly explained. How the diagnosis of psoriasis was diagnosed (clinical examination only, histopathology etc..). Which data were collected from the patients? Among the medical history, a relevat data to my opinion is to analyze the influence of therapeutics for psoriasis on cardiovascular comorbidies
3) results: the authors should adequately explain the demographic characteristics of the patient cohort: age, sex, previous treatmens for psoriasis, other comorbidities that could be also associated with heart disease etc...
4) results: I do not clearly understand why the authors present three cases of patients who required treatment for cardiovascular disease separately (this is not a case series, so this section is not needed to my opinion)
5) the figure of erythrodermic psoriasis is not needed for the scope of this study
6) tables are not adequately explained. for example, table 1 is too long. it may be better to present only cumulative data.. how many males and females; average data; how many patients with dislipidemia etc.. rather than enumarate all the data from a single patient (in this case i would suggest showing these data as supplementary files).
3)
Author Response
Thank you for reviewing it.
Please find my response below.
- abstract is too long. The authors should separate the various sections providing a summary of the methods, results and conclusions.
→I made it short.
- the methods are not clearly explained. How the diagnosis of psoriasis was diagnosed (clinical examination only, histopathology etc..). Which data were collected from the patients? Among the medical history, a relevat data to my opinion is to analyze the influence of therapeutics for psoriasis on cardiovascular comorbidies
→Added evidence of diagnosis in dermatology and rheumatology (CASPAR). I would like to discuss the effect of treatment history in the future.
- results: the authors should adequately explain the demographic characteristics of the patient cohort: age, sex, previous treatmens for psoriasis, other comorbidities that could be also associated with heart disease etc...
→I have added the results of statistical analysis for complications, co-morbidities, and gender. The analysis of previous treatment and CCTA abnormality was not done in this study, so it will be a future issue.
- results: I do not clearly understand why the authors present three cases of patients who required treatment for cardiovascular disease separately (this is not a case series, so this section is not needed to my opinion)
→It has been removed from the main text and made supplementary data.
- the figure of erythrodermic psoriasis is not needed for the scope of this study
→It has been removed from the main text and made supplementary data.
- tables are not adequately explained. for example, table 1 is too long. it may be better to present only cumulative data.. how many males and females; average data; how many patients with dislipidemia etc.. rather than enumarate all the data from a single patient (in this case i would suggest showing these data as supplementary files).
→Table 1 has been changed for simplicity and clarity. I have decided to attach the original Table 1 as supplemental data.
Please find attached the corrected text.
